# Average Intensity of Low-Frequency Sound and Its Fluctuations in a Shallow Sea with a Range-Dependent Random Impedance of the Liquid Bottom

**Fengqin Zhu** [1,2]🄳, **Oleg E. Gulin** [3,*]🄳 **and Igor O. Yaroshchuk** [3]

[1] College of Underwater Acoustic Engineering, Harbin Engineering University, Harbin 150001, China; zhufq@gdou.edu.cn
[2] Key Laboratory of Underwater Acoustic Technology, Harbin Engineering University, Harbin 150001, China
[3] V.I. Il'ichev Pacific Oceanological Institute, Far-East Branch of Russian Academy of Sciences, 690068 Vladivostok, Russia; yaroshchuk@poi.dvo.ru
[*] Correspondence: gulinoe@poi.dvo.ru; Tel.: +7-423-231-2617



**Featured Application: The results obtained in the present research show the complex nature of the propagation of sound signals and noise in the conditions of the shallow waters of the Arctic basin shelf. Despite the rather simple hydrology of the water column, a high degree of diversity of the characteristics of the sub-bottom layer was observed here, and a statistical approach was applied to describe the influence of this layer. Analysis of the fluctuation nature of the signal indicates the need to search for new approaches to processing information received by acoustic measuring systems in the Arctic seas. The quantitative estimates obtained by the authors of the degree of influence of random bottom parameters in typical shallow-water Arctic regions can be used to predict propagation losses of low and medium-frequency sound signals, which is important in underwater communications, detection problems and environmental issues of reducing the impact of anthropogenic signals and noise on marine mammals. The statistical analysis performed in this paper is useful for predicting the presence of bottom areas with gas-saturated sediments. In addition, pronounced fluctuations in the parameters of the sub-bottom layer often make the known deterministic approaches to solving inverse problems of reconstructing the characteristics of the medium from the measured acoustic field inadequate in the Arctic seas and require the development of statistical methods.**

**Abstract:** In this study, the problem of the influence of a horizontally inhomogeneous liquid bottom impedance, given by random Gaussian function of the speed of sound and by density, on the propagation of low-frequency sound in a shallow-water waveguide is considered. The model parameters are referenced to the conditions of sound propagation in the regions of the seas of the Russian Arctic shelf. By the example of statistical modeling of the sound field intensity, we show that sound speed fluctuations in the bottom lead to similar effects that were previously established for volumetric fluctuations of the speed of sound in the water layer. With the distance from the source, the decrease in the average intensity slows down in comparison with a deterministic medium in which there are no fluctuations. This deceleration of the decay of the intensity in a random waveguide can be significant already at short distances. Changes in the law of decay of intensity at a fixed frequency are mainly determined by the correlation radius of inhomogeneities and the average penetrability of the bottom, which leads to attenuation of sound propagating in the waveguide.

**Keywords:** random media; computational shallow-water acoustics; range-dependent waveguides; local modes; inhomogeneous impedance of the bottom; statistical modeling and computing

## 1. Introduction

Many works [1–8] (see also numerous references therein) are devoted to the propagation of sound signals in the Arctic Ocean. One review [1] of the current state of research indicates that in the post-war period until the early 1990s, acoustic research was mainly of a military orientation, associated with the navigation and detection of nuclear submarines. Since the 1990s, the attention of researchers has switched to the study of climatic changes in the Arctic due to the rapidly deteriorating state of the ice cover (the extent and thickness of sea ice have sharply decreased over the past 50 years), due to an increase in the concentration of carbon dioxide and other greenhouse gases in the atmosphere [2–4]. Acoustic tomography of the ocean water column has been developed [5,6], which makes it possible, by means of remote sensing of the water masses of the Arctic basin, to study their temperature dynamics in a year-round mode (acoustic thermometry) [7,8]. Both in the case of military and thermometry purposes, experimental and theoretical studies were carried out for long-distance routes (2000–2700 km) of the central part of the deep Arctic Ocean [1,7,8], namely, experiments TAP, 1994; ACOUS, 1998–1999; and CAATEX, 2019–2020. They pursued the aim of identifying the features of the propagation of ultra-low frequency signals (tens of hertz) in the presence of ice cover and stratification (as a rule, in the form of a near-surface sound channel), characteristic of the central part of the Arctic basin. As a result, it became possible to remotely register the temperature rise in the Arctic using acoustic methods, to track the movement of large water masses flowing from the Atlantic and the Pacific Ocean to the Arctic Ocean and their transformation, to develop multipurpose acoustic systems with inclusion in the integrated observing system for the Arctic region [1,7].

However, the features of the sound fields formed on the long paths of the deep part of the Arctic Ocean, which, owing to the above-mentioned works, have been well studied, are very different from the features of sound propagation in the shallow zones (with typical depths up to 100 m) of the Arctic basin shelf. Here, from the acoustical point of view, of greater interest is not the study of changes in stratification and ice cover (which may not exist at all, for example, in August–September), but the influence of the inhomogeneous bottom structure. In shallow seas, as is known, it is the nature of the bottom that dominates the propagation of sound signals. The present work refers specifically to the study of the features of low-frequency sound signal propagation in the conditions of the shallow seas of the Arctic shelf. The importance of such studies is due to the solution of the problems of prospecting and developing mineral deposits and related environmental issues, particularly the impact of anthropogenic signals and noise on marine mammals living in the shallow seas of the shelf. From open sources, only a few works are known, devoted to the propagation of acoustic signals in the shallow Arctic seas adjacent to the land. Thus, in articles [9,10], in a deterministic formulation, the propagation of low-frequency sound (hundreds of hertz) for several short paths (5 km long) of the Kara Sea shelf with an inhomogeneous bottom is considered. From an acoustic point of view, sound propagation in the Arctic shelf waters occurs under conditions of a shallow waveguide with a nearly uniform distribution of sound velocity over depth and very diverse properties of bottom sediments. Such conditions, as indicated above, differ significantly from the conditions of signal propagation in the waveguide of the central Arctic (the effect of the bottom there is of little significance, and if taken into account, then fragmentarily and phenomenologically [7,8], its actual parameters are unknown).

In [9], on the basis of experimental geophysical data, it is indicated that the upper layer of bottom sediments on the Arctic shelf comprises, as a rule, unconsolidated or weakly consolidated sediments, which are characterized by the presence of various structures. These are layered series of deposits with a length up to tens of kilometers, narrow vertical channels related to natural gas emissions with craters at the water–bottom interface. In the composition of the material of these structures, there is an alternation of areas of frozen, thawed and gas-saturated sediments [11], with a significant

dispersion of the values of the speed of sound $c_1$ and density $\rho_1$. Thus, the upper part of the sedimentary layer of the Arctic shelf bottom is a medium with a substantially inhomogeneous spatial distribution of the main acoustic parameters $c_1$ and $\rho_1$, which describe the impedance of the lower boundary of the waveguide. For the most adequate description, this indicates the need to consider the bottom impedance as a random spatial function and studying the statistical problem of sound propagation in a homogeneous waveguide of a shallow sea with horizontal fluctuations of the bottom impedance along the path. This paper presents the results of a statistical analysis of such a model problem for the characteristic values of the waveguide and bottom parameters, corresponding to the known 3D seismic data [9], obtained in the water area of the Kara Sea shelf. A statistical analysis of this problem, which assumes coverage of a wide range of possible values of the speed of sound in the upper layer of bottom sediments, makes it possible to draw fairly representative conclusions about the average characteristics of the acoustic field not only for the Kara Sea but also for the propagation of sound in the shallow waters of the Russian Arctic shelf as a whole.

Note that the statistical problem of the influence of the fluctuating bottom impedance on the propagation of a sound signal in a 2D inhomogeneous shallow-water waveguide is apparently formulated and solved for the first time. From a fundamental point of view, of interest are the regularities of the behavior of the average intensity, which describe the transmission losses of the signal in a random medium waveguide bottom sediments, and its fluctuations revealed in the work. The applied significance of the results of this work lies in the quantitative estimates obtained by the authors of the degree of influence of random bottom parameters in typical Arctic regions, which can be used to predict transmission loss of low and medium-frequency signals in these regions and areas with similar conditions. It is also important to note that the statistical analysis performed in this work is useful for predicting the presence of bottom regions with gas-saturated sediments. This is of interest, on the one hand, for the exploration of minerals in the regions of the Arctic shelf, and on the other hand, for the indication of possible places of emission of greenhouse gases into the atmosphere.

## 2. Statement of the Problem and Solution Representation

A monochromatic sound field of circular frequency $\omega$ in a shallow sea range-dependent waveguide is described by linear equations of acoustics with suitable boundary conditions, which are set on the basis of the continuity of the sound pressure and velocity components when crossing the interfaces. For media with losses, it is also assumed that the condition of limiting absorption is satisfied. At a constant density $\rho$ in water for the acoustic pressure $p$, the acoustic equations are reduced to the Helmholtz equation of the form (implicit time factor $e^{-i\omega t}$ is assumed):

$$\left( r^{-1}\frac{\partial}{\partial r}\left(r\frac{\partial}{\partial r}\right) + \frac{\partial^2}{\partial z^2} + \frac{\omega^2}{c^2} \right) p(r,z) = -\frac{\delta(r)\delta(z-z_0)}{2\pi r}, \tag{1}$$

where $(r,z)$ are the coordinates of the cylindrical system, $r$ is directed horizontally and the point radiation source is located at the point ($r = 0$, $z = z_0$) (axially symmetric formulation of the problem), and $c$ is the speed of sound in water. The boundary condition on the sea surface $p(r,0) = 0$, and the condition at the bottom ($z = H$) corresponds to the continuity of pressure and the velocity component that is normal to the boundary $H$. In the wave zone of the source, the field $p(r,z)$ is sought using the expansion in local modes of an irregular waveguide:

$$p(r,z) = \sum_m G_m(r)\varphi_m(r,z); \quad \frac{\partial^2}{\partial z^2}\varphi_m(r,z) + \left[k^2(r,z) - \kappa_m^2(r)\right]\varphi_m(r,z) = 0 \tag{2}$$

In Equation (2), $k = \omega/c$, $\kappa_m(r)$ are the eigenvalues, $m = 1, 2, \ldots$ , $\varphi_m$ are the eigenfunctions of the Sturm–Liouville problem, which, on the surface and at the bottom of the ocean, satisfy the following boundary conditions ($\varphi'_m = \partial\varphi_m(r,z)/\partial z$):

$$\varphi_m(r,0) = 0, \quad \varphi_m(r,H) + g_m(r)\varphi'_m(r,H) = 0 \tag{3}$$

The function $g_m(r)$ in Equation (3) characterizes the impedance of the penetrable bottom, and it is a random function due to fluctuations in the speed of sound $c_1$ and density $\rho_1$ in the bottom sediments. This circumstance carries novelty to the mathematical formulation of the boundary value problem since the influence of random fluctuations of the bottom impedance in the horizontal direction on the propagation of sound in the water layer has not been studied previously. Thus, it is obvious from Equations (2) and (3) that the eigenfunctions and eigenvalues, and with them the local modes of the waveguide, will also be random functions of $r$. It can be shown that in an irregular waveguide in the forward scattering approximation, the amplitudes of modes $G_m(r)$ are determined by the following solution presented in matrix form:

$$G(r) = \{G_m(r)\} = A(r)exp\left\{\int_0^r \left[i\kappa(\xi) - \left(\kappa(\xi)V(\xi)\kappa^{-1}(\xi) - V^T(\xi)\right)/2\right]d\xi\right\}b(0), \tag{4}$$

where $\kappa(r)$ is the diagonal matrix of eigenvalues $\{\kappa_m(r)\}$, $\kappa_m \cdot r \gg 1$, $A(r) = (i/8\pi r)^{1/2}\kappa^{-1/2}(r)\kappa^{-1/2}(0)$; $b(0) = \{\varphi_m(0,z_0)\kappa_m^{1/2}(0)\}$ is the column vector of the initial amplitudes of modes; and exp {...} is the matrix exponential. In Equation (4), $V(r)$ is a matrix with elements $V_{mn}(r) = \int_0^\infty \frac{\varphi_m(r,z)}{\rho(r,z)}\frac{\partial\varphi_n(r,z)}{\partial r}dz$, and $V^T(r)$ is a transposed matrix $V$. The latter matrices describe mode coupling due to horizontal changes caused by fluctuations of the speed of sound and density in the bottom sediments. The analytical matrix form of solution (4) has a number of advantages over the recording of the solution for individual mode amplitudes, which was used earlier in [12–16]. First, it can be seen from the matrix form of solution (4) that all the new effects arising from the propagation of sound in a randomly inhomogeneous medium are determined by the matrix exponential (as a function of $r$). Therefore, these effects (changes in the laws of decay of sound intensity with distance, or in the signal transmission losses) quantitatively weakly depend on the horizons of the location of the radiation point ($z_0$) and the observation point ($z$). Secondly, the form of solution (4) is optimal in terms of calculations. The matrix exponential does not contain additional parameters, such as the initial amplitudes of modes $\varphi_m(0,z_0)$ in [12–16]. In addition, no special computational schemes are required to compute matrix exponential in (4). At the same time, such schemes are necessary when solving differential equations for the mode amplitudes in the framework of the well-known approximations, such as Wentzel–Kramers–Brillouin (WKB) and the mode parabolic equation (MPE). Note that solution (4) for mode amplitudes corresponds to the approximation of a one-way propagation (OW) [17] for boundary value problem (2)–(3) to the original Equation (1). If we neglect the backscattered field, then this OW approximation directly follows from the causal equations of the first order, derived previously. They are equivalent to the original boundary Equations (1)–(3). The causal equations in question were obtained by the method of differentiation with respect to the parameter (imbedding method) in the works [12,14,18,19] for a 2D-inhomogeneous marine environment and in [20] for 3D inhomogeneities. The imbedding method for wave problems is well described in [21].

Solution (4) takes into account the scattering of modes at any angles, but not exceeding $90°$. In what follows, neglecting backscattering, we call it the exact solution, or OW. If the density of the medium does not change in the horizontal direction, then the matrix $V(r)$ in Equation (4) becomes skew-symmetric: $V_{mn}(r) = -V_{nm}(r)$, $V_{nn} = 0$. If inhomogeneities of the medium change smoothly and the scattering angles are small, so that $\kappa(r)V(r)\kappa^{-1}(r) \approx V(r)$, then the WKB approximation in the horizontal direction can be obtained from Equation

(4), as well as the approximation of the MPE [10,17,22,23]. As shown by our calculations, the WKB and MPE solutions for the waveguide models considered below also provide a sufficient approximation to the exact solution.

By calculating the pressure field $p(r,z)$ according to Equations (2)–(4) for each random realization $c_1(z, r)$, $\rho_1$ from an ensemble of $N$ realizations, it is easy to obtain the change in the average intensity, or the average function of transmission loss for sound propagation along the path in a randomly inhomogeneous waveguide:

$$\langle I \rangle = \left\langle |p|^2 \right\rangle = \sum_n \left\langle |G_n|^2 |\varphi_n|^2 \right\rangle + \sum_{(n \neq m)} \left\langle G_n G_m^* (\varphi_n \varphi_m^*) \right\rangle \tag{5}$$

where angled brackets denote statistical averaging. Similarly, according to well-known formulas, other statistical characteristics of the field and intensity are calculated. In this work, we particularly need an expression for the second normalized statistical moment $S = (\langle I^2 \rangle - \langle I \rangle^2)^{1/2}/\langle I \rangle$ ($S^2$ is known as the scintillation index), which describes the intensity fluctuations during sound propagation in the waveguide.

## 3. Model of a Stochastic Waveguide

To carry out a numerical analysis, we further reference the parameters that, according to [9,10], are characteristic of the shelf zones of the Russian Arctic seas, particularly the Kara Sea (in the absence of ice cover on the surface in summer). It is important to note that the presence of ice cover is insignificant for the purposes of our study. This study is associated with the influence of fluctuations of parameters in bottom sediments. It is well known that the presence of ice on the sea surface leads to a faster decay of the signal during propagation [1,4,7]. As a result, the transmission loss curves will be located lower than without considering the presence of ice on the surface. We consider a shallow-water waveguide with depths of 30 m and 40 m with a horizontal surface and bottom, having homogeneous profiles of sound speed $c = 1460$ m/s and density $\rho = 1$ g/cm$^3$, in which a tonal sound signal with a frequency of 250 Hz propagates. The bottom is an absorbing liquid half-space of unconsolidated sediments with a refractive index at the water–bottom interface $n = (c/c_1)(1 + i\beta_1)$, $\beta_1 = 0.02$ (see Figure 1). The parameters of the bottom sediments, $c_1(z,r)$, $\rho_1(r)$, randomly vary along the signal path. In this case, the impedance function $g_m(r)$ in the boundary condition (3) is determined by the local values of the comparison waveguides: $g_m(r) = -i\rho_1(r)\rho^{-1}[k^2 - \kappa_m^2(r)]^{-1/2}$. In this work, following the measurement data given in [9,10], we put $\rho_1(r) = \langle \rho_1 \rangle = 1.85$ g/cm$^3$. We also took into account the fact that, as calculations show, random variations in density $\delta\rho_1(r)$, $\rho_1(r) = \langle \rho_1 \rangle + \delta\rho_1(r)$, have a much weaker effect on sound propagation than fluctuations in the speed of sound. This fact is well known from the theory (see, for example, [17]). The variations in density in bottom sediments can be neglected if not too low radiation frequencies are examined ($f = 2\pi\omega > 1$ Hz) and there are no large-amplitude jumps of $|\delta\rho_1/\langle \rho_1 \rangle|$ in the medium.

For the speed of sound in the bottom sediments, we first considered the random process $c_1(r) = \langle c_1 \rangle + \delta c_1(r)$ (see Figure 2), setting it with a Gaussian probability distribution with an exponential correlation function: $B_{c1}(r_2 - r_1) = \sigma_{c1}^2 \exp(-|r_2 - r_1|/L_r)$. Based on the processing data [9,10], as well as information from [24], the characteristic scale $L_r$ of the inhomogeneities vary is further chosen to be 1 km, and the intensity of fluctuations $\sigma_{c1}^2 = \langle (\delta c_1/\langle c_1 \rangle)^2 \rangle = 1.7 \cdot 10^{-3}$ (corresponds to the amplitude $|\delta c_1| \approx 60$ m/s). Note that although the amplitude of fluctuations is small, $\sigma_{c1} \ll 1$, for bottom sediments, it is more than an order of magnitude higher than the values used in modeling the fluctuations of the speed of sound in water, which are caused, for example, by background internal waves [22,24,25].

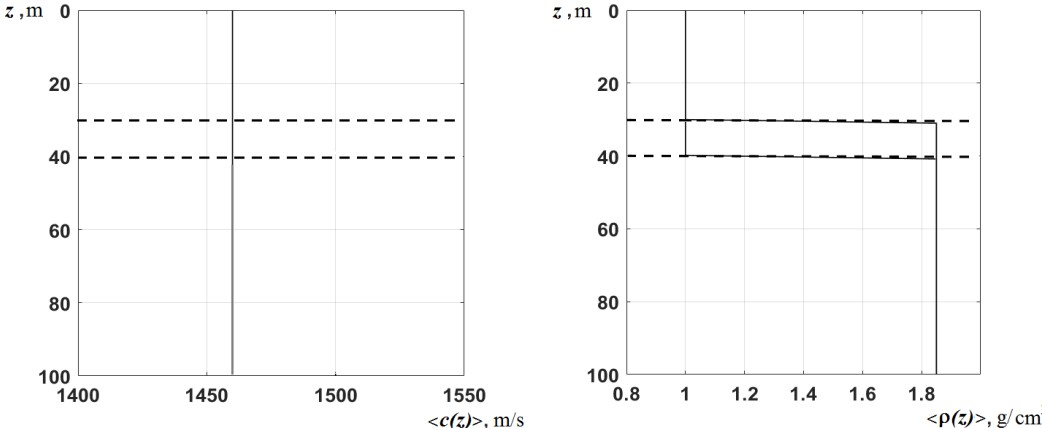

**Figure 1.** Average profiles of the speed of sound $\langle c(z) \rangle = \langle c_1(z) \rangle$ = 1460 m/s and density $\langle \rho(z) \rangle$ in the waveguide with the liquid bottom. Dotted lines show the bottom boundary location for two examples.

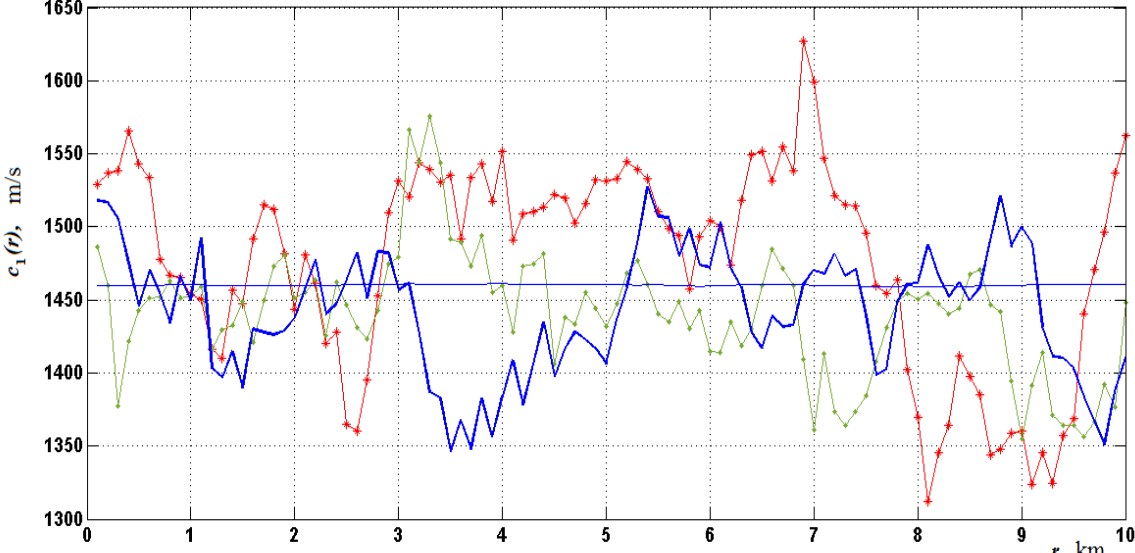

**Figure 2.** Three arbitrary random realizations of the speed of sound in the bottom half-space from the ensemble of *N* realizations, $L_r$ = 1 km.

From Figure 2, it is seen that the signal propagating along the path passes through the bottom areas with inhomogeneities of the "soft" ($c_1 < c$) and "rigid" ($c_1 > c$) types, which is typical for the shelf of the Arctic seas. In this case, inhomogeneities of the "soft" type with $c_1 < c$ (including $c_1 << c$) are due to the presence of gas saturation in sediments [26]. In this case, when $c_1 < c$, even in the absence of absorption $\beta_1$, there are no propagating (trapped) modes in the waveguide, and all modes are to be leaky. In the second case, for $c_1 > c$, depending on the degree of "rigidness" of the bottom, the first several modes at a frequency of 250 Hz are propagating and weakly attenuating (for $\beta_1 \neq 0$). To find the local eigenvalues $\kappa_m(r)$ and eigenfunctions $\varphi_m(r)$, the reference to the Pekeris cut on the complex plane of $\kappa_m$ was carried out. Therefore, the required number of propagating and leaky modes was taken into account in the sum (2) [17,27]. As a rule, with the waveguide hydrology and sound frequency described above, it was sufficient to use no more than 4–6 modes of different types in the calculations for the purpose of our statistical modeling. It should be emphasized that despite the uniform nature of the sound velocity and density profiles in the water layer, the random waveguide as a whole, being a water–bottom sediment fluctuating system, is not the traditional Pekeris model [27]. This is obviously a much more complex 2D waveguide of a shallow sea (see also Section 5).

## 4. Statistical Analysis of the Sound Transmission Loss in a Shallow Sea: The Presence of Bottom Sediments with Fluctuating $c_1(r)$

Figures 3 and 4 contain the results of statistical modeling of the average intensity (5) in a shallow-water waveguide with a highly penetrable bottom (Figures 1 and 2). Average intensity curves are given in dB relative to the intensity level in the free field at a distance of 1 m from the source. In this case, as was shown in [16,28] using the example of volume fluctuations of the speed of sound in the water layer, one should expect the greatest statistical effect during the sound signal propagation. This is because the wavenumbers of even the lowest (small grazing angles) modes along the path can have a relatively large imaginary part (at $c_1 \approx c$), which is significantly influenced by random inhomogeneities. Over the course of the numerical simulation, an ensemble of realizations $N$ from 300 to 1000 was considered. It is clearly seen from Figures 3 and 4 that fluctuations in the bottom impedance, as well as fluctuations in the speed of sound in the water layer [16,28], led to a significant slowdown in the average intensity decrease along the path, which is well noticeable even for small distances. For example, in the presence of fluctuations in the speed of sound in the bottom sediments for a waveguide with the depth of 30 m, the level of average intensity at a distance of 3 km is 10 dB higher than for a deterministic (when $\delta c_1 = 0$) waveguide, and at a distance of 5 km, this difference is about 20 dB. However, in the situation under consideration, this effect is noticeably less pronounced than in the presence of fluctuations in the speed of sound in the water column $\delta c$, despite the relative smallness of such $\delta c$. In the latter case, as was shown in [28], the slowing down of the average intensity decay at a distance of 10 km can reach 100 dB (for the frequency of 500 Hz and $H = 50$ m). This fact is of a general nature [24], since inhomogeneities in the bottom sediments have a weaker effect on the values of the modal wave numbers of the waveguide $\kappa_m(r)$, particularly on the imaginary parts $\kappa_m(r)$ describing acoustic energy losses. The modal wavenumbers $\kappa_m(r)$ determine the fluctuations of the signal intensity in individual realizations and, therefore, the statistical effect of transmission loss reduction, shown in Figures 3 and 4. In terms of physics, the fact of a decrease in the transmission loss of sound can be explained by the appearance of local fluctuation waveguides along the propagation path, with varying degrees of focusing of the acoustic energy. The degree of focusing by these local fluctuation waveguides is determined by fluctuations of the imaginary parts of the eigenvalues $\kappa_m(r)$. The larger the value of these imaginary parts $\text{Im}(\kappa_m(r))$, the more they fluctuate, and in the statistical mean, there is a decrease in the transmission loss of a signal. It is seen from Figure 4 that an increase in the waveguide depth with the same values of other parameters weakens the above-described effect of slowing down the decay of the average intensity. This is explained by the decrease in the imaginary parts of $\kappa_m(r)$ of the lowest-number modes with increasing depth. It is these modes that determine the attenuation of the acoustic field in the sea with increasing distance from the source (compare the deterministic curves in Figures 3 and 4).

It should be noted that in the situations examined, the adiabatic approximation describes the average transmission loss well. The curves in Figures 3 and 4 for OW and "adiabatic" differ by fractions of a decibel, so they are hardly distinguishable on the graphs. This is due to the overwhelming contribution of the first mode in most realizations. With distance from the source, this mode is the least attenuated, and the necessary condition for the adiabatic approximation, $L_r \gg |\kappa_1(r) - \kappa_n(r)|^{-1}$, is satisfied. Nevertheless, in individual intensity realizations, the result of mode coupling can be noticeably manifested. First of all, this refers to those realizations that describe a sufficiently fast decay of the signal intensity along the propagation path.

In Figures 3 and 4, it is clearly seen that the curves of statistical modeling at $r > 1.5$–2 km are quite smooth. This is due to two circumstances. The first is the statistical averaging of the intensity over the ensemble of realizations. Second, the step $\Delta r$ of calculating the mode amplitudes by formula (4) was chosen much less than the characteristic scale (correlation radius) of inhomogeneities $\Delta r \ll L_r$. At the same time, it was comparable to the scale of the interference structure of the field $\Delta r \sim |\kappa_m(r) - \kappa_n(r)|^{-1}$.

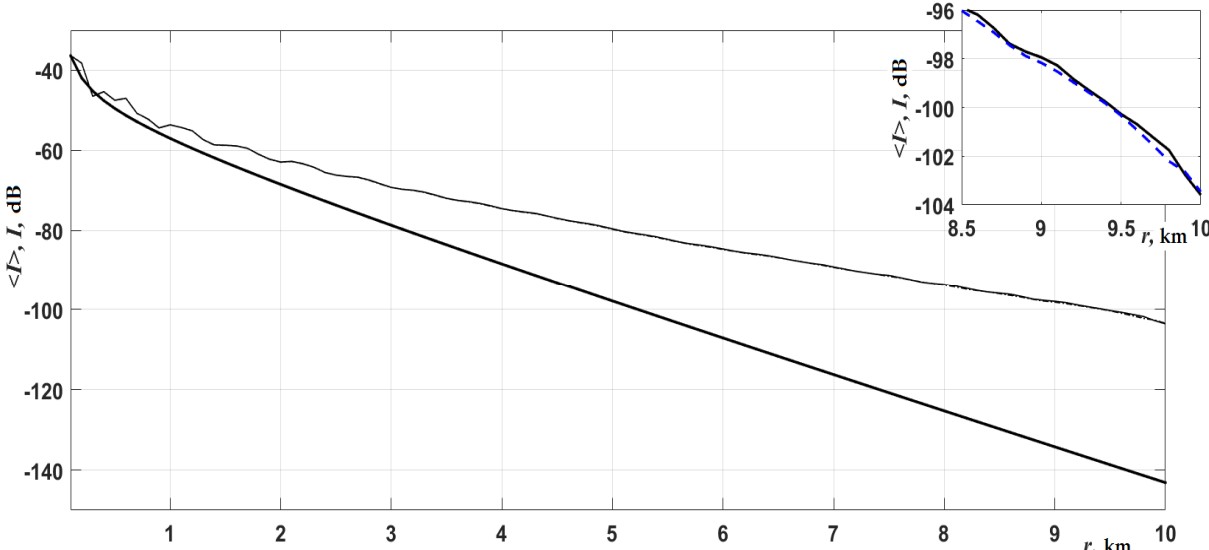

**Figure 3.** The average intensity (with the opposite sign it is transmission loss) in the waveguide with the random impedance. $N = 10^3$. $H = 30$ m, $z = z_0 = 14$ m. The upper solid and dashed curves, poorly distinguishable from each other, correspond to the OW solution and adiabatic approximation ($V_{mn} = 0$). For a distance of 8.5–10 km, these curves are shown in the inset in the upper right corner of the figure. The lower bold curve is a deterministic solution ($\delta c_1 = 0$), averaged over spatial oscillations.

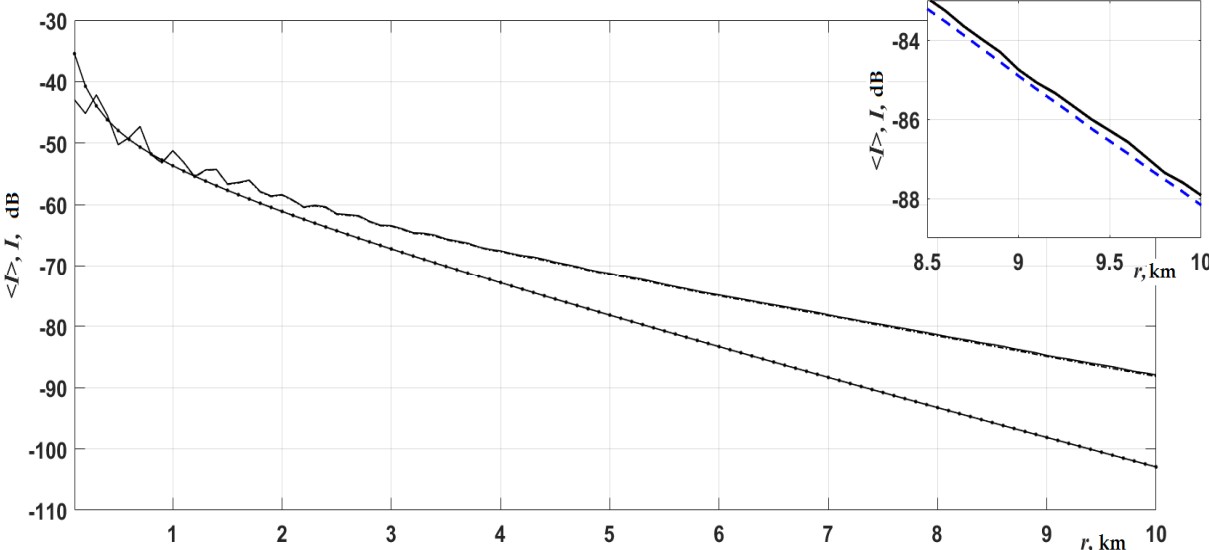

**Figure 4.** Curves of the average intensity are similar to Figure 3. For a distance of 8.5–10 km, two upper curves are shown in the inset in the upper right corner of the figure. The lower marker curve is a deterministic solution. $N = 10^3$. $H = 40$ m, $z = z_0 = 16$ m.

Figure 5 shows a graph of intensity fluctuations $S$ (scintillation index) along the signal propagation path in the considered randomly inhomogeneous waveguide of a shallow sea. It is clearly seen that fluctuations of the intensity develop rather rapidly. Already at distances of 1.5–2 km from the source, $S$ begins to noticeably exceed 1, which means the appearance of strong intensity fluctuations in the waveguide, which continue to increase with a distance without reaching the saturation regime. From the same distances $r > 1.5$–2 km in Figures 3 and 4, there is a divergence of the curves corresponding to the statistical and deterministic dependences, which indicates a decrease in loss of the average intensity in the shallow-water waveguide. In the presence of random inhomogeneities in

the water layer of the shallow sea, this fact of the appearance of strong fluctuations with increasing distance is known from [24,25,29] and was also noted in [16,30]. The appearance of strong intensity fluctuations in the waveguide means an increase in the spread of the intensity levels as a function of the distance [30] in individual realizations. For example, the range of this spread in intensity levels for the situation presented in Figures 3 and 5 at a distance of 5 km from the source is 53 dB, and at a distance of 10 km, it reaches 103 dB.

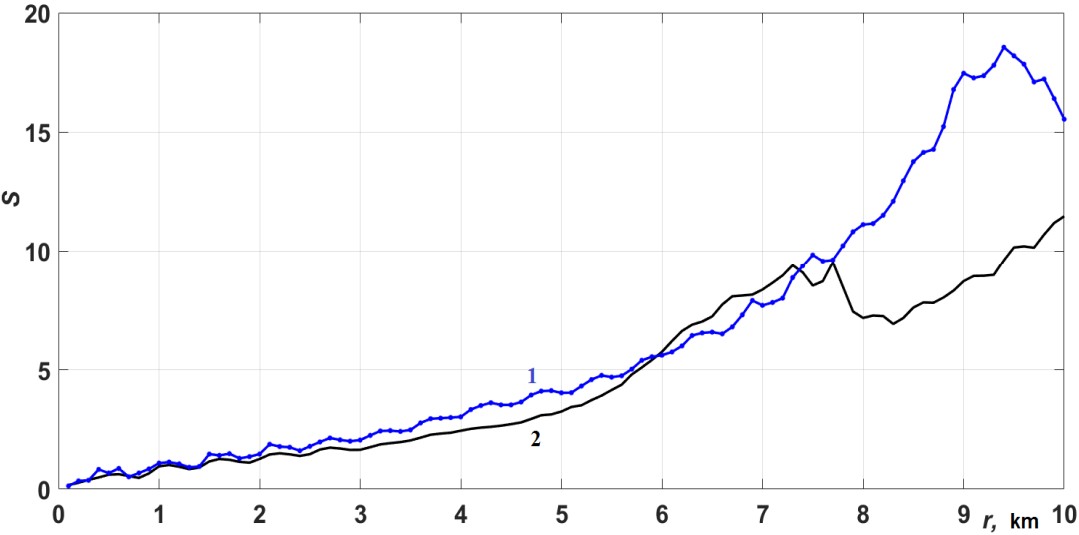

**Figure 5.** Development of intensity fluctuations $S$ in stochastic waveguides according to the OW solution (2)–(5). $H = 30$ m, $z = z_0 = 14$ m. Marker curve 1 corresponds to the waveguide with random $c_1(r)$. Solid black curve 2 corresponds to the waveguide with random $c_1(z,r)$.

Above, a model of liquid bottom sediments with a horizontal scale of sound velocity fluctuations $L_r = 1$ km was considered. The scale of fluctuations in depth $L_z$ was assumed to be large enough (theoretically, infinite). It is also possible to consider randomly stratified bottom sediments with a limited $L_z$ value. Qualitatively, the pattern of sound propagation in the water column changes little. The quantitative differences in the intensity decay in this case are considered in the next section.

## 5. Statistical Analysis of Sound Transmission Loss in the Waveguide: Stratified Bottom Sediments with Random $c_1(z,r)$

For the speed of sound in the bottom sediments, consider a two-dimensional Gaussian random field $c_1(z,r) = \langle c_1 \rangle + \delta c_1(z,r)$ with an exponential correlation function: $B_{c1}(z_2 - z_1, r_2 - r_1) = \sigma_{c1}^2 \exp(-|z_2 - z_1|/L_z - |r_2 - r_1|/L_r)$. Let us assume that in the vertical direction, fluctuations occur in the bottom sedimentary layer 15 m thick (Figure 6), and below, there is a half-space with $c_1 = c = 1460$ m/s. This thickness of the bottom sedimentary layer was not chosen by chance. For the used frequency of 250 Hz, more than two wavelengths fit in the sedimentary layer, which, as shown by our calculations and the results of works [9,10,26], neutralizes the influence of sedimentary rocks occurring at greater depth. The characteristic scale of variation of the inhomogeneities over depth is chosen to be $L_z = 10$ m, and the intensity of fluctuations $\sigma_{c1}^2$ is given by the previously defined value. In this case, as is clear from Figure 6, even the local comparison waveguides (in each section along the path of the original 2D waveguide) are not Pekeris waveguides, and the calculation of the eigenvalues and eigenfunctions is significantly complicated due to the appearance of the so-called "jumping" (or "jumped out") poles on the complex plane of $\kappa$ [9]. Comparison of the curves in Figure 7 shows that the presence of random stratification in sediments with a finite correlation radius $L_z$ somewhat lowers the level of average intensity with a distance. Thus, at distances of 8–10 km, the difference between the levels of the two upper curves is 3–5 dB. Respectively, the scintillation index shown by

curve 2 in Figure 5 for random stratification of bottom sediments with $L_z = 10$ m grows more slowly than in the case of a large scale $L_z$. However, the main effect of slowing down the average intensity decay or reducing the transmission loss remains actual.

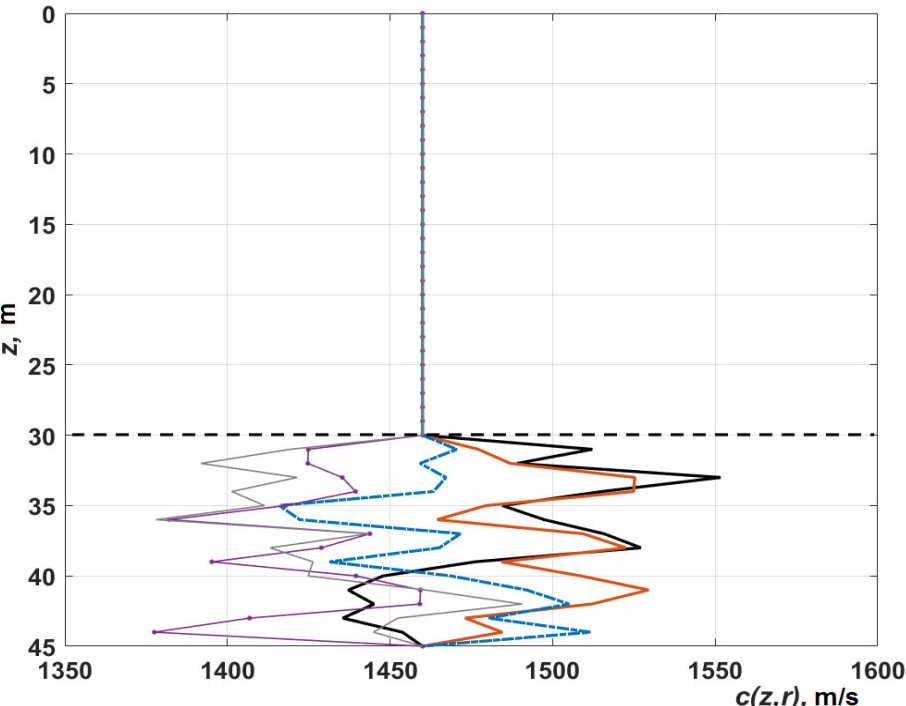

**Figure 6.** An example of the speed of sound profile in the water and in liquid bottom sediments for five arbitrary random realizations at some distance $r$ from the source. Dashed line at $z = 30$ m shows the water–bottom interface.

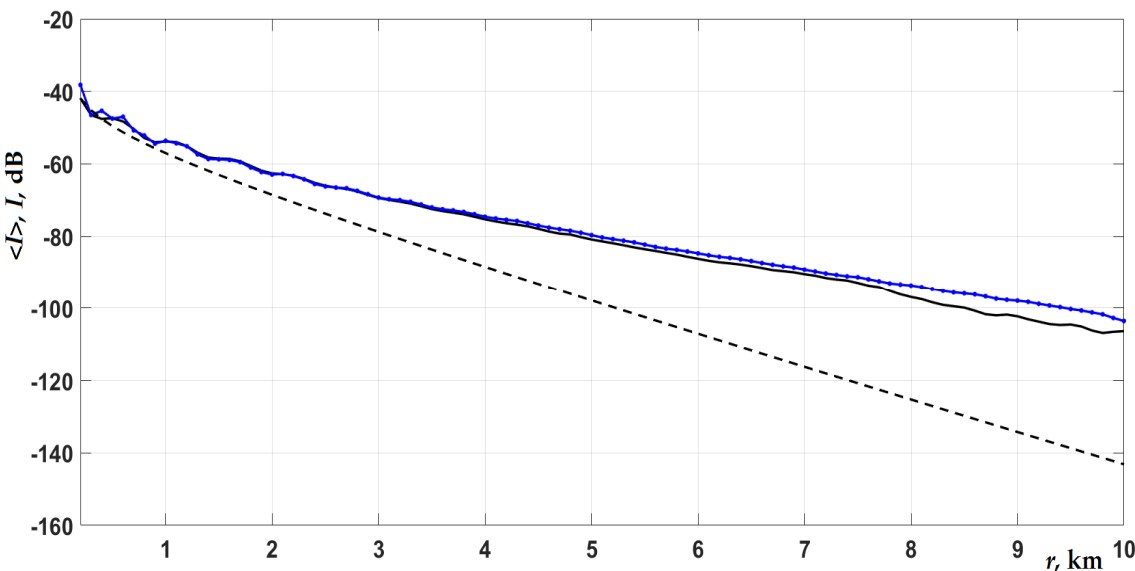

**Figure 7.** Comparison of the average intensity $\langle I \rangle$ in waveguides with a random field $c_1(z,r)$ and $c_1(r)$. $H = 30$ m, $z = z_0 = 14$ m. The marker curve corresponds to a waveguide with a random process $c_1(r)$ (Figure 3), the solid black curve corresponds to a waveguide with a random field $c_1(z,r)$ (Figure 6). These curves describe the OW solution (2)–(5). The lower curve is a deterministic solution (see Figure 3).

## 6. Model Calculations

The solution of the eigenvalue problem (2)–(3) and the subsequent calculations of the acoustic intensity, the results of which are presented in this work, were carried out on the basis of expressions (4)–(5) using program codes developed by the authors in the MATLAB environment. We preliminarily performed the generation of random processes $c_1(r)$ and fields $c_1(z,r)$. To perform this procedure, the correlation matrix $D$ of the random column vector $\eta$ was represented as the product $D = Q \cdot Q^T$, where $Q$ is the lower triangular matrix. Then the value of the random vector $\eta$ can be obtained by the formula $\eta = Q\mu$, where $\mu = [\mu_1, \mu_2, \mu_3 \ldots \mu_n]^T$ is a vector composed of independent Gaussian random variables with the properties $\langle \mu_j \rangle = 0$, $\langle \mu_j^2 \rangle = 1$. Note that the procedure for calculating the matrix $Q$ is called Cholesky decomposition [31] and it is presented in the MATLAB standard software package. This procedure was applied not only to the one-dimensional process $c_1(r)$, but also to the two-dimensional field $c_1(z,r)$ after the discretization with respect to $r$ and $z$, determined by the conditions $(r_{j+1} - r_j) \ll L_r$ and $(z_{j+1} - z_j) \ll L_z$. We also note that to simulate Gaussian random fields, some authors use a rather economical but approximate computational procedure for randomizing the spectral representation of fields, the advantages of which rise with an increase in the dimension of the random field [32,33]. However, in all the calculations presented in this work, we managed to restrict ourselves to the approach described above.

In the case of a waveguide with random stratified sediments, investigated in Section 5, the calculation of eigenvalues and eigenfunctions was significantly complicated due to the appearance of the so-called "jumping" poles [9] in the complex plane of horizontal wavenumbers $\kappa$. As a rule, they correspond to leaky modes, a significant part of the energy of which is localized in the sedimentary layer, where random waveguide structures are formed for a propagating sound signal in the certain number of realizations (see Figure 6). The authors' programs for searching for complex eigenvalues based on the algorithm of the wave impedance method [12,24,34], as well as the known program "Kraken" [17,35], are not very suitable for obtaining even the initial approximation to such $\kappa_n$. Therefore, in practice, when searching for eigenvalues, we had to use the procedure of sequential division of regions on the complex plane of horizontal wavenumbers $\kappa$, which greatly slowed down the calculations.

## 7. Discussion

In this paper, we considered the statistical problem of the behavior of the sound intensity of a low-frequency point source in a shallow-water homogeneous waveguide with randomly inhomogeneous liquid sediments of the bottom. The performed statistical modeling has shown that fluctuations in the speed of sound in the bottom sediments can lead to the effects of the appearance of strong fluctuations in the signal intensity at small distances from the source and a slowdown in the decay of the average intensity in the waveguide. Similar effects were previously established for random inhomogeneities in the speed of sound in the water layer with thermocline [16], which are usually caused by internal waves. In relation to the shelf zones of the Arctic basin, however, the nearly homogeneous speed of sound over the sea depth is typical, which reduces the relevance of studying the effect of the mechanism of internal waves on the propagation of sound in the shallow Arctic seas. At the same time, the diversity of the impedance characteristics of the bottom in this region and the significant lack of data on these characteristics translate the acoustic problem of signal propagation in the Arctic shelf into a statistical problem. In this formulation, the statistical analysis of the effect of fluctuations in the speed of sound in bottom sediments is partly a study that reflects the degree of our ignorance of the spatial varies in the bottom parameters at the waveguide regions where the sound signal propagates. The presented results of statistical modeling predict expectations regarding the intensity levels of the propagating signal in a shallow-water waveguide with a highly penetrable (in average) bottom. They clearly demonstrate the fast stochastization of the signal in such a waveguide with a simultaneous weakening of its intensity attenuation,

which is associated with a decrease (in the statistical sense) in the leakage of acoustic energy into the bottom half-space. Just as in the case of fluctuations of the speed of sound in water, the magnitude of the described effects is determined by the average penetrability of the waveguide bottom and the horizontal scale $L_r$ of the sediment inhomogeneities. The more reflective ("rigid" or "soft") the bottom is at a fixed $L_r$, the less pronounced the described statistical effects become. The same is true for a decreasing $L_r$ value for given parameters characterizing the degree of bottom rigidity. In terms of physics, the fact of a decrease in the transmission loss of sound can be explained by the appearance of local fluctuation waveguides along the propagation path, with varying degrees of focusing acoustic energy. The degree of focusing by these local fluctuation waveguides is determined by fluctuations of the imaginary parts of the eigenvalues $\kappa_m(r)$. The larger the value of these imaginary parts $\kappa_m(r)$, the more they fluctuate, and in the statistical mean, there is a decrease in the transmission loss of a signal. Here, we should refer to [36], in which the possibility of forming a fluctuation waveguide channeling the wave energy of a point source along random layers is theoretically shown for a random half-space of a layered medium.

In works [29,37–42] for the statistical analysis of the influence of random inhomogeneities in water on the propagation of sound in a marine environment, a theory was developed that was called the diffusion approximation. It is based on assumptions that allow describing the effect of internal waves on acoustic fields in a deep ocean with an underwater sound channel. For a shallow sea, the main concern is to consistently take into account the sound losses caused by the bottom, and in the simulation, it is necessary to correctly calculate the random complex eigenvalues and eigenfunctions of the modes. In this situation, the indicated theory is unsuitable for an adequate study of the influence of sound speed fluctuations both in the water column and in bottom sediments on the propagation of acoustic signal, since one of the conditions for the applicability of the diffusion approximation for a horizontally inhomogeneous sea is: $\mathrm{Im}^{-1}(\kappa_m(r)) \gg L_r$. Additionally, the more penetrable to acoustic modes the bottom is, the greater the discrepancy between the results of the diffusion approximation and the exact simulation (see, for example, [43]) due to the growth of the imaginary part of the eigenvalues in random realizations. For the models considered in this paper, in many random realizations the inequality $\mathrm{Im}^{-1}(\kappa_m(r)) < L_r$ is fulfilled. Thus, our results complement those of well-known works [37–42], providing an analysis of wave statistics in a new region of characteristic values of the problem parameters.

In this study, we analyzed the influence of Gaussian random fluctuations of the speed of sound in the bottom of a shallow sea. However, by analogy with the analysis carried out in [16], it can be assumed that for non-Gaussian fluctuations of the speed of sound in bottom sediments, the results obtained will remain valid.

## 8. Conclusions

In this work, we formulated and studied the statistical problem of the propagation of a low-frequency sound signal in a homogeneous water column of a random shallow-water waveguide with 2D Gaussian sound speed fluctuations in the layer (and half-space) of liquid bottom sediments. Based on the performed statistical modeling, the following main results were obtained.

1. For waveguides with an essential average penetrability of the bottom boundary, the fact of a significant slowing down of the decay of the average signal intensity (reduction of transmission loss) along the propagation path has been established. Compared to a deterministic waveguide with similar regular parameters, the reduction in transmission loss in the presence of fluctuations can reach several tens of decibels (in the average statistical sense). This reduction in signal transmission loss occurs already on rather short paths of 3–10 km.
2. Simultaneously with the decrease in transmission losses, there is an increase in the fluctuations of the signal intensity. We found that the scintillation index describing the development of such fluctuations grows rather rapidly with distance, exceeding

unity level already at distances of several kilometers. Thus, the stochastization of a signal in the randomly inhomogeneous waveguide occurs rather quickly. An increase in the scintillation index is observed along the entire propagation path without a transition to the saturation regime, which is typical for media with energy losses.

3.   The maximum effect of decreasing the average transmission loss was achieved at a large value of the vertical correlation radius of the sound velocity fluctuations $L_z$ in bottom sediments. However, even with a small value of $L_z \sim \lambda$, where $\lambda$ is the sound wavelength, the result of reducing the average loss in the waveguide remains in effect.

4.   Numerical simulations show that the results obtained are described fairly well by the adiabatic approximation, which neglects the coupling of modes. This fact, as well as random fluctuations of the speed of sound in the water layer, is explained by the overwhelming contribution to the intensity of most realizations of the first (least attenuated) mode, which manifests itself already at rather small distances from the source.

**Author Contributions:** F.Z., data curation, writing—original draft, funding acquisition; O.E.G., conceptualization, formal analysis, validation, writing—review and editing, methodology; I.O.Y., supervision, project administration, software, investigation. All authors have read and agreed to the published version of the manuscript.

**Funding:** This research was funded by the National Natural Science Foundation of China (grant no. 41406041). The research was also carried out as a part of the Russian State assignment on the topic "Study of the fundamental basis of the origin, development, transformation, and interaction of hydroacoustic, hydrophysical, and geophysical fields of the World Ocean" (state number registration: AAAA-A20-120021990003-3).

**Institutional Review Board Statement:** Not applicable.

**Informed Consent Statement:** Not applicable.

**Data Availability Statement:** This study did not report any data.

**Conflicts of Interest:** The authors declare no conflict of interest.

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
