# Peer review of "Average Intensity of Low-Frequency Sound and Its Fluctuations in a Shallow Sea with a Range-Dependent Random Impedance of the Liquid Bottom"

_applsci, doi:10.3390/app112311575_

Round 1

Reviewer 1 Report

Please to kindly see the attached file.

Thank you.

Reviewer 2 Report

The authors present a well-organized and well-founded work. They propose a particular analysis of the influence of the fluctuation of the speed of sound in the water column and in the bottom sediments on the propagation of the acoustic signal in the shallow Arctic seas. The authors justify why it is not appropriate to use the diffusion approximation theory in this study. The conclusions are consistent with the results of your model. Comments that can be made to improve the work are not relevant.

Author Response

The authors are grateful to the reviewer for his attention to the work.

Reviewer 3 Report

Summary:
A statical analysis of a model problem, obtained in the water area of the Kara Sea shelf is presented. It is shown that the sound-speed fluctuations in the bottom lead to similar outcomes that were established for volumetric fluctuations of the speed of sound in the water layer. Moreover, the results are confirmed by quantitative estimates of random bottom parameters in typical Arctic regions. This can be used to predict transmission loss of low and medium- frequency signals with similar conditions in the same regions.

 Suggestions:
1.  Introduction section should have different paragraphs to divide different subjects for easier comprehension.

2. Please reread the text again, there are a few grammar errors, such as:

a) Line 81, differ?

b)Line 152-153-154, should be rephrased also check the grammar

c) Line 160

3. In figures, you can use legends, for better visualization and shorter caption.

4. Conclusion section can be an individual section to emphasize more on the achievements and not wrapped around the discussion.

Round 2

Reviewer 3 Report

Thanks for addressing the feedback.

This manuscript is a resubmission of an earlier submission. The following is a list of the peer review reports and author responses from that submission.

Round 1

Reviewer 1 Report

Please to see the attached file. Thank you,

Reviewer 2 Report

This is a review of the corrected manuscript "Average intensity of low-frequency sound and its fluctuations in a shallow sea with a range-dependent random impedance of the liquid bottom" submitted to Applied Sciences.

The paper is in this state quite well written and the methods employed seem relevant to me. I find that the authors have responded adequately to the concerns of the three previous reviewers and thus improved the manuscript. Overall, the results are interesting and I think the manuscript can be published as is. The quality of the figure and the way they are described may still be improved but it is not an obstacle to the understanding of the results.

Author Response

The authors are grateful to the reviewer for a useful and objective consideration of the work.

Reviewer 3 Report

The main content of this work is to simulate the intensity fluctuation in the Pekeris waveguide with a random fluid bottom in the Arctic. The quality of the writing is not bad while the manuscript was not kindly written.

The biggest problem of this work is that no originality/novelty has been found in this manuscript. While the authors claims that this work is related to the Arctic, most of this work has been performed by the modeling based on an simple ocean environment with constant sound speed profile and fluid random bottom. For such environment, the behavior of intensity fluctuation has been well understood by past theoretical studies mainly done by American and Russian scientists, even if this study is not. 

These days, the sound modeling in the Arctic will be only valuable with the comparison of the experimental data in the Arctic. Also, realistic modeling considering the ice and ocean sediment physics will be interesting in the community. At present, the strength of this work is so weak.

Round 2

Reviewer 3 Report

Thank you for the kind responses for the authors.

But, the reviewer is still wondering that this manuscript needs improvement. The main contribution of this manuscript is only to calculate Eq. (5) in the Arctic environment numerically. A formulation for 2D random bottom is not derived in this work. Eqs. (1) and (4) is the contribution of previously published works of the authors (The reviewer think that Ref. [4-6] are absolutely good works ). 

And, a new numerical algorithm is not described in the work. The authors shortly mention to overcome the difficult of eigenvalue searching, which was immediately observed in the past work of the authors. 

Finally, the simulation results may be valuable in the sonar detection but the results of the authors must need to verify with those of other modeling technique such as RAM or the precise analysis with the experimental measurements.